# Therapeutic Activity of a Topical Formulation Containing 8-Hydroxyquinoline for Cutaneous Leishmaniasis

**DOI:** 10.3390/pharmaceutics15112602

**Published:** 2023-11-08

**Authors:** Sarah Kymberly Santos de Lima, Ítalo Novaes Cavallone, Dolores Remedios Serrano, Brayan J. Anaya, Aikaterini Lalatsa, Márcia Dalastra Laurenti, João Henrique Ghilardi Lago, Dalete Christine da Silva Souza, Gabriela Pustiglione Marinsek, Beatriz Soares Lopes, Renata de Britto Mari, Luiz Felipe Domingues Passero

**Affiliations:** 1Institute of Biosciences, São Paulo State University (UNESP), Praça Infante Dom Henrique, s/n, São Vicente 11330-900, SP, Brazil; sarahkslima@gmail.com (S.K.S.d.L.); id.cavallone@gmail.com (Í.N.C.); gabriela.marinsek@unesp.br (G.P.M.); bs.lopes@unesp.br (B.S.L.); renata.mari@unesp.br (R.d.B.M.); 2Laboratory of Pathology of Infectious Diseases (LIM50), Department of Pathology, Medical School, São Paulo University, São Paulo 01246-903, SP, Brazil; mdlauren@usp.br; 3Department of Pharmaceutics and Food Science, Faculty of Pharmacy, Universidad Complutense of Madrid, Plaza Ramon y Cajal s/n, 28040 Madrid, Spain; drserran@ucm.es (D.R.S.); branaya@ucm.es (B.J.A.); 4CRUK Formulation Unit, Institute of Pharmacy and Biomedical Sciences, University of Strathclyde, 161 Cathedral Street, Glasgow G4 0RE, UK; aikaterini.lalatsa@strath.ac.uk; 5Center for Natural and Human Science (CCNH), Federal University of ABC, Santo André, São Paulo 09210-580, SP, Brazil; joaohglago@gmail.com (J.H.G.L.); christinesilvax@gmail.com (D.C.d.S.S.); 6Institute for Advanced Studies of Ocean, São Paulo State University (UNESP), Rua João Francisco Bensdorp, 1178, São Vicente 11350-011, SP, Brazil

**Keywords:** topical treatment, 8-hydroxyquinoline, quinolines, *L. (L.) amazonensis*, cutaneous leishmaniasis

## Abstract

Cutaneous leishmaniasis exhibits a wide spectrum of clinical manifestations; however, only a limited number of drugs are available and include Glucantime^®^ and amphotericin B, which induce unacceptable side effects in patients, limiting their use. Thus, there is an urgent demand to develop a treatment for leishmaniasis. Recently, it was demonstrated that 8-hydroxyquinoline (8-HQ) showed significant leishmanicidal effects in vitro and in vivo. Based on that, this work aimed to develop a topical formulation containing 8-HQ and assess its activity in experimental cutaneous leishmaniasis. 8-HQ was formulated using a Beeler base at 1 and 2% and showed an emulsion size with a D_50_ of 25 and 51.3 µm, respectively, with a shear-thinning rheological behaviour. The creams were able to permeate artificial Strat-M membranes and excised porcine skin without causing any morphological changes in the porcine skin or murine skin tested. In BALB/c mice infected with *L. (L.) amazonensis*, topical treatment with creams containing 1 or 2% of 8-HQ was found to reduce the parasite burden and lesion size compared to infected controls with comparable efficacy to Glucantime^®^ (50 mg/kg) administered at the site of the cutaneous lesion. In the histological section of the skin from infected controls, a diffuse inflammatory infiltrate with many heavily infected macrophages that were associated with areas of necrosis was observed. On the other hand, animals treated with both creams showed only moderate inflammatory infiltrate, characterised by few infected macrophages, while tissue necrosis was not observed. These histological characteristics in topically treated animals were associated with an increase in the amount of IFN-γ and a reduction in IL-4 levels. The topical use of 8-HQ was active in decreasing tissue parasitism and should therefore be considered an interesting alternative directed to the treatment of leishmaniasis, considering that this type of treatment is non-invasive, painless, and, importantly, does not require hospitalisation, improving patient compliance by allowing the treatment to be conducted.

## 1. Introduction

Natural or synthetic quinolines are a group of bioactive compounds with high chemical versatility and diverse pharmacological activities [1,2]. The 8-hydroxyquinoline (8-HQ) molecule is a subclass of quinoline, that is considered a chelating agent, a property associated with most bioactivities of this compound [3,4]. Consequently, interest in 8-HQ has grown exponentially in the last two decades, as this molecule has shown a wide variety of pharmacological activities such as neuroprotection, anticancer, antimicrobial, and antiparasitic agents [5,6].

As an antiparasitic agent, 8-HQ was already demonstrated to be active on *Trypanosoma cruzi* epimastigotes, the aetiologic agent of Chagas disease, and showed similar efficacy to Nifurtimox, a reference drug used in the treatment of Chagas disease [7]. In sleep sickness, 8-HQ derivatives were active in blood forms of *Trypanosoma brucei*, showing selective indexes between 11 and 48, which were superior in comparison to the reference drug. Furthermore, all derivatives were also observed to interact with DNA and eliminate parasites by oxidative stress [8].

Furthermore, male MF1 mice infected with *Schistosoma mansoni* were found to show a reduction in liver parasitism of 65% after oral 8-HQ administration. Histologically, a decrease in the volume of liver granulomas was observed in treated animals, and an increase in anti-*S. mansoni* IgG antibodies was observed. Comparatively, praziquantel, a reference drug used in the treatment of human schistosomiasis, reduced the parasite burden by 62%, but did not reduce the volume of liver granulomas or elevate the levels of anti-*S. mansoni* IgG antibodies, suggesting that immunomodulatory activity could have helped the antiparasitic activity of 8-HQ [9].

In leishmaniasis, several studies have shown that 8-HQ and its derivatives were active on promastigotes and intracellular amastigotes of *L. (L.) amazonensis*, *L. (L.) infantum*, and *L. (L.) braziliensis*, and that activity was higher than amphotericin B, a reference drug used in the treatment of leishmaniasis [10]. Furthermore, it was shown that such a molecule was able to depolarise the mitochondrial membrane, leading to the death of leishmania parasites. Further studies also verified that BALB/c mice infected with *L. (L.) infantum* or *L. (L.) amazonensis* submitted to subcutaneous treatment with 8-HQ and clioquinol showed a significant reduction in the size of cutaneous lesions and the number of tissue parasites; furthermore, the efficacy of 8-HQ in a mouse model of cutaneous leishmaniasis was associated with an increase in the Th1 immune response [11,12]. Importantly, 8-HQ and clioquinol induced no liver or renal damage in experimental animals [13], suggesting that 8-HQ is an interesting molecule to develop a new and effective drug directed to leishmaniasis.

Recently, 8-HQ showed a wide range of leishmanicidal activity on promastigotes and intracellular amastigotes of *L. (L.) amazonensis*, *L. (L.) infantum chagasi*, *L. (V.) guyanensis*, *L. (V.) naiffi*, *L. (V.) lainsoni*, and *L. (V.) shawi*, demonstrating multispectral activity and higher selectivity than miltefosine. This is likely attributed to the leishmanicidal activity of 8-HQ and the production of nitric oxide (NO) in macrophages. Furthermore, the administration of 8-HQ by an intralesional route at 10 and 20 mg/kg caused a significant reduction in parasite burden in the skin of BALB/c mice infected with a strain of *L. (L.) amazonensis* that was isolated from a patient with anergic diffuse leishmaniasis. Additionally, these animals produced higher levels of IFN-γ and lower levels of IL-4 cytokines, indicating an immunomodulatory response stimulated by 8-HQ during leishmania infection [14].

Taken together, these studies indicated that 8-HQ had higher bioactivity in *Leishmania* sp., which is more selective than drugs conventionally used in the treatment of leishmaniasis, suggesting that 8-HQ is indeed a hit alternative drug. It is of the utmost importance to characterise new treatments for both visceral (VL) and cutaneous leishmaniasis (CL), considering the existence of a limited arsenal of chemotherapeutic possibilities that are based on pentavalent antimonials, amphotericin B, and miltefosine.

Pentavalent antimonials (Sb^v^) were recommended by the WHO as the first-line drug for the treatment of leishmaniasis [15], and can be administered in doses of 20 mg/kg^−1^ subcutaneously for 28–30 days [16,17]. However, antimonials showed variable responses to disease (35% to 95%); in addition, this treatment is limited, considering all side effects that are induced in patients, including cardio and hepatotoxicity [18,19]. Amphotericin B is recommended as the second-line drug for the treatment of leishmaniasis, mainly in regions where patients are resistant to antimony [20]. Administration is given daily or on alternate days with 1 mg/kg^−1^ during 15–20 intravenous infusions, resulting in high efficacy [21]. Despite its high efficacy, the treatment causes significant side effects, including fever, hypokalaemia, and nephrotoxicity; in addition, the treatment is costly and requires prolonged hospitalisation [22]. Lipid formulations of amphotericin B were developed to minimise the severe side effects of amphotericin B [23,24], but they are still expensive, primarily for low-income countries [25,26]. Miltefosine is an alkyl phosphocholine derivative with antineoplastic activity and a significant leishmanicidal effect. It is the only drug administered orally that is currently approved for the treatment of leishmaniasis [27]. Treatment is given for 28 days at variable doses, depending on geographic location and the clinical form of leishmaniasis. The main problem associated with miltefosine treatment is the long half-life in the human body that can cause gastrointestinal toxicity; teratogenicity is another concern related to the treatment. Furthermore, the cure rate is variable depending on the infecting species, but a lack of efficacy has also been described in human infections [28,29], suggesting the existence of parasites resistant to miltefosine [30].

Despite the pharmacological effect of such drugs on leishmaniasis, it is important to note that none of them was originally developed to treat human leishmaniasis, so the current treatment is based on repurposed drugs whose toxic effects in patients remain unacceptable while remaining highly invasive and requiring hospitalisation [31]. In this case, the development of a topical therapy would represent an important milestone in the treatment of cutaneous leishmaniasis; however, until now, there has been no effective topical treatment. Taking into account the scarcity of topical treatments, this work aims to develop a low-cost cream containing 8-HQ, a compound with a multispectral effect on *Leishmania* species, and to evaluate its preclinical efficacy in an experimental cutaneous leishmaniasis model. Prepared creams exhibited significant rheological, physical, and permeability features, while no toxic events were observed in the skin of BALB/c treated mice. 8-HQ creams were able to reduce the size of the lesion and tissue parasitism in experimental cutaneous leishmaniasis. Our data support that 8-HQ topical formulations can be extemporaneous readily translatable low-cost topical treatments for cutaneous leishmaniasis that can be easily translated into industrially scalable topical therapies for cutaneous leishmaniasis.

## 2. Materials and Methods

### 2.1. Materials

RPMI 1640 medium (Thermo Scientific, Waltham, MA, USA) was supplemented with 10% inactivated foetal bovine serum (Thermo Scientific), 1% pyruvate, 1% of non-essential amino acids, 10 μg/mL of gentamicin and 1000  U/mL penicillin, and 0.1% of 2-mercaptoethanol (R10). Schneider medium (Sigma-Aldrich, St. Louis, MO, USA) was supplemented with 10% heat-inactivated foetal bovine serum (FBS), 10 μg/mL of gentamicin, and 1000 U/mL of penicillin (S10). The 8-hydroxyquinoline molecule 8-hydroxyquinoline (purity > 99%) was obtained from Sigma-Aldrich (USA). Glucantime^®^ was obtained from Sanofi-Aventis (São Paulo, Brazil).

### 2.2. Preparation of Creams Containing 8-HQ

To prepare the Beeler’s base (10 g), firstly, the oily phase was prepared by melting cetyl alcohol (1.5 g) and white wax (0.1 g) (Sigma) in a water bath at 70 °C. Subsequently, butylhydroxytoluene (2 mg) was added as the oily phase. The aqueous phase, composed of sodium lauryl sulphate (0.2 g), propylene glycol (1 g), and distilled water (q.s. 10 g), was then heated up to the same temperature as the oil phase and mixed with the oily components to produce the Beeler base cream. 8-HQ was dispersed in 50 µL of propylene glycol at 1% (0.1 g) or 2% (0.2 g) prior to being incorporated into the Beeler’s base using a mortar and pestle. The creams were packaged in 30 mL amber glass bottles and no changes in the physical appearance of the creams were observed for six months at room temperature (~25 °C).

### 2.3. Physical Characterisation of Creams

Samples of all creams were diluted in deionised water (1:100 *w*/*w*), and 300 µL aliquots of the diluted creams were added to the integrated cuvette of the laser particle analyser with a detection range between 1 and 135,000 μm (Microtrac S3500, Microtrac, PA, USA) to determine the particle size and distribution of 8-HQ creams. The size and distribution of the prepared creams were expressed by the median diameter of the volume (MV) and the D_10_, D_50,_ and D_90_ (indicating the percentages of particles having 10%, 50%, and 90% of the diameter equal to or lower than the given value) [32]. Measurements were performed in triplicate.

### 2.4. Viscosity Analysis

The viscosity of the creams prepared was evaluated in triplicate using a Brookfield Rheometer (Middleborough, MA, USA), model DV-III, fitted with a temperature control probe. A 5 cm cone–plate measuring geometry was used (Spindle CP-42). The temperature of all measurements was maintained at 25 °C. Viscosity (cP) and shear stress (D × cm^−2^) were determined over a speed rate from 0 to 0.5 rpm and from 0.5 to 0 rpm, and a shear rate from 0 to 1.92 (1/s). Before measurements, a standard of 30,000 cp was analysed. The fluidity parameter was calculated using linear regression from the slope of the shear stress versus the shear rate plot.

### 2.5. Permeation Studies with Creams Containing 8-HQ

A Strat-M^®^ artificial membrane (Transdermal Diffusion Test, 25 mm, Sigma-Aldrich, Madrid, Spain) with 0.33 mm thickness, engineered to mimic human skin, was initially used. This multi-layer artificial membrane possesses a tight top layer coated with a lipid blend resembling the lipid chemistry of the human stratum corneum (SC) and a porous lower layer resembling the epidermis and dermis layers [33]. This membrane possesses equivalency to human skin for the skin permeation of many drugs and claims to have better correlations compared with other biological membranes [34]. Strat-M^®^ was mounted between the donor and receptor chamber of Franz diffusion cells with a diffusion area of 1.766 cm^2^ and a cell volume of 12 mL capacity. The receptor chamber was filled with a mixture of phosphate buffer (pH 5.5) and methanol (1: 1, *v*:*v*). The donor compartment was filled with PBS for 30 min until the system reached 35 °C. The PBS in the receptor chamber was removed and 100 mg of creams containing 1% or 2% 8-HQ were added to the donor compartment in strict contact with the membrane. Samples of 1 mL were withdrawn from the receptor chamber at the following time points, 5, 10, 15, 30, 45, 60, 120, 240, and 360 min, and samples were analysed using HPLC, as described below. The receptor chamber was immediately replenished with pre-warmed buffer. Cumulative amounts of 8-HQ permeated through the artificial membrane were analysed at 360 min. Additionally, permeability studies were performed using porcine skin from the ear of 3-month-old male pigs obtained from the slaughterhouse. The skin (thickness: 1.66 ± 0.16 mm) was prepared as previously described [35]. At the end of each permeability study (360 min), the skin samples were wiped with an ethanol-impregnated cotton bud to remove the excess formulation. Skin samples were cut in half and one half was fixed using 4% paraformaldehyde (pH 7) for histological studies, while the other half was weighed and homogenised with 2 mL of PBS pH 7.4 buffer, diluted 1:2 with methanol, vortexed, and centrifuged (10 min, 10,000 rpm) [32]. The supernatant was analysed for HPLC.

The collected samples were analysed using UHPLC (Ultimate 3000 standard Quaternary System). The integration of the peaks was performed using the program Chromeleon 7.3.1.6535). An Analytic Kinetex EVO 5 mm C18 reverse phase HPLC column (150 × 4.6 mm) was used for analysis. Isocratic elution was used with a mobile phase consisting of acetonitrile and methanol (52:48). The flow rate was set at 1 mL/min, and the injection volume was 10 μL. Detection was carried out at 246 nm, the retention time was detected in 2.25 min of run, and a linear calibration curve was achieved between 0.38 and 400 μg/mL^−1^ (R^2^ > 0.9906).

A standard curve was constructed with 8-HQ and regression analysis was used to calculate the slopes and intercepts of the linear portion of each graph. The steady-state flux (JSS), the permeability coefficient (P), the diffusion coefficient, and the lag time were estimated as previously described by Lalatsa et al. [36].

Each formulation was tested in triplicate. Regression analysis was used to calculate the slopes and intercepts of the linear portion of each graph, and the following equations were applied to each formulation.

To calculate the steady-state flux,
(1)$$jss=\frac{dC}{dX}\times A$$
where *jss* is the steady-state flux (μg/cm^2^/h), *dC*/*dX* is the amount of 8-HQ permeating the membrane over time (μg/h), and *A* is the surface area of contact of the formulation.

To calculate the permeability coefficient (*P*), the following Equation (2) was employed:(2)P=jss/cd
where *cd* is amount of drug applied in the donor compartment

The diffusion coefficient (μm/h) was calculated by using Equation (3):(3)$$jss=\frac{D\times k}{h}\;cd$$
where *h* is the thickness of the membrane (μm^2^) [36].

### 2.6. Animals

Six- to eight-week-old male BALB/c mice were obtained from the Animal Facility Center of the Medical School of São Paulo University, housed in accordance with Animal Welfare Committee standards, and had access to food and water ad libitum throughout the study under a 12 h light cycle. This study was carried out following the recommendations of the Guide for the Care and Use of Laboratory Animals of the Brazilian National Council of Animal Experimentation. The protocol was approved by the Committee on the Ethics of Animal Experiments of the Institutional Animal Care and Use Committee at the Medical School of São Paulo University (CEP1648/2022). For all experimental procedures, the animals were anaesthetised with ketamine (100 mg/kg) and xylazine (10 mg/kg).

### 2.7. Cutaneous Toxicity Studies with Creams Containing 8-HQ

Twenty five healthy male BALB/c mice were divided into five groups: group 1 (G1) constituted control mice; group 2 (G2) constituted animals treated with 1.7 mg of blank Beeler’s base cream by the topical route at the base of the tail, with a surface area approximately of 5.2 mm^2^; groups 3 (G3) and 4 (G4) constituted animals treated with 1.7 mg of cream containing 1% or 2 of 8-HQ by the topical route at the base of the tail, respectively; group 5 (G5) was treated with 50 mg/kg of Glucantime^®^ by the intralesional route. Animals were treated during 14 consecutive days, once daily. The physical conditions of the animals were monitored once a week. One week after the last dose, the animals were euthanised with a lethal dose of thiopental. Skin fragments and inguinal lymph nodes were collected to perform different assays. All animals survived till the end of the study.

### 2.8. Parasite

The *L. (L.) amazonensis* parasite (MHOM/BR/73/M2269) was identified using monoclonal antibodies and isoenzyme electrophoretic profiles at the Leishmaniasis Laboratory of the Evandro Chagas Institute (Belém, PA, Brazil). Parasite species were grown in S10 medium at 25 °C. In all experiments, the parasites were in a stationary phase of growth and were in the first passage of culture to perform in vivo experiments.

### 2.9. Infection and Experimental Treatment

Thirty male BALB/c mice were subcutaneously infected at the base of the tail with 10^6^ promastigote forms of *L. (L.) amazonensis* in the stationary phase of growth. Five animals received only sodium chloride 0.9% (*w*/*v*) under the same route (healthy group). Four weeks after infection, *L. (L.) amazonensis*-infected BALB/c mice were divided into six groups: group 1 (G1) and group 2 (G2) constituted infected animals treated with 1.7 mg of cream containing 1% or 2 of 8-HQ by the topical route at the base of the tail, with a surface area approximately of 5.2 mm^2^, respectively; group 3 (G3) was treated only with Beeler’s base cream by the topical route; group 4 (G4) was treated with 50 mg/kg of Glucantime^®^ by the intralesional route; group 5 (G5) was the infected control; and group 6 (G6) constituted non-infected, non-treated animals. Treatment was started in the fifth week post-infection (PI), and the animals were treated for 14 consecutive days, once daily. The physical conditions of the animals were monitored once a week. One week after the last dose, the animals were euthanised with a lethal dose of thiopental. Skin fragments and inguinal lymph nodes were collected to perform different assays. All animals survived the infection till the end of the study.

### 2.10. Development of Lesions and Determination of Tissue Parasitism

The clinical course of the development of the lesion was assessed once a week with the aid of a digital micrometer with range of 0–25 mm and 0.001 mm accuracy (Mitutoyo, Aurora, IL, USA). The skin parasite load was determined using the quantitative limiting dilution assay. Briefly, a skin fragment from the base of the tail and inguinal lymph nodes was collected aseptically, weighed, and homogenised in S10 medium. The tissue suspensions were subjected to 12 serial dilutions with four replicate wells. The number of viable parasites was determined based on the highest dilution in which the promastigotes could grow after 10 days of incubation at 25 °C.

### 2.11. Cytokine Production Studies

Inguinal lymph nodes and skin fragments from different groups were collected aseptically, weighted, macerated in R10 medium, and centrifuged at 10,000 rpm, 6 min, 4 °C. The supernatants were collected, and the amounts of IL-4 and IFN-γ (ThermoFischer, Waltham, MA, USA) were quantified by sandwich enzyme-linked immunosorbent assay (ELISA) according to the manufacturer’s recommendations. The concentrations of cytokines were normalised by the weight of each organ collected.

### 2.12. Statistical Analysis

All experiments were repeated at least three times and the values obtained were expressed as mean ± standard error. Statistical analyses were performed using GraphPad Prism 5.0 software and the ANOVA test was used to rate differences between groups. Statistical significance was established at *p* < 0.05.

## 3. Results

### 3.1. Measurement of Particle Size

Physical characterisation demonstrated that the distribution of all creams was arranged in such a way that 10% of the particles (D10) had a size of ~10 µm (Figure 1). Beeler’s base and the cream containing 1% of 8-HQ exhibited a D_50_ of 25 µm; however, the cream containing 2% of 8-HQ showed a significant increase in D_50_ (51.27 µm) in comparison with Beeler’s base and cream 1%. Beeler’s base and 1% cream showed a similar D_90_, with a particle size of 92.84 and 89.93 µm, respectively, but the cream 2% presented the highest D_90_ (146.77 µm) in comparison to all creams.

### 3.2. Viscosity

No statistical differences were observed in the viscosity values between the blank Beeler’s base cream and creams containing 1 or 2% of 8-HQ at any of the rpm tested (Figure 2A). All three creams showed a shear-thinning rheological behaviour (Figure 2B), referring to the decrease in shear stress at higher shear rates. This behaviour is optimal for topical pharmaceutical products, since upon application on the skin the extensibility improves, allowing an easier application and, hence, better patient compliance.

### 3.3. Franz Cell Diffusion Assay

Permeation studies were performed with 1% and 2% 8-HQ creams using porcine skin, as well as with artificial membranes (Strat-M membrane) engineered to mimic human skin (Table 1). Steady-state flux was observed to be 1.57 higher in porcine skin incubated with 1% cream than with 2% cream (*p* < 0.05). However, in artificial membranes, the steady-state flux was similar when incubated with creams of 1 or 2%.

In contrast, the permeability and diffusion coefficient parameters were 3.33 and 3.18 times higher in the skin incubated with the cream of 1% than with the cream of 2%, respectively. On the other hand, artificial membranes incubated with creams of 1 or 2% did not show significant differences between such parameters. The latency time parameters (lag time) were similar in the skin and artificial membranes incubated with 1 or 2% creams.

After 360 min of incubation, porcine skin fragments were collected and fixed in buffered formalin (pH7.4). In this regard, creams containing 1% (Figure 3A) or 2% (Figure 3B) of 8-HQ did not change the morphology of porcine skin, considering that the epidermis, superficial and deep dermis, glands, hair follicles, and capillaries were preserved.

### 3.4. Histological Changes in the Skin from Healthy BALB/c Mice Treated with Topical Creams Containing 8-HQ

The histological sections of the skin of healthy animals without treatment showed a normal morphology of the epidermis and dermis, as well as of the glands, hair follicles, and capillaries (Figure 4A). Similarly, the skin of animals treated with blank Beeler’s base (Figure 4B) or cream containing 1% (Figure 4C) or 2% (Figure 4D) of 8-HQ showed a similar morphology to the control group (Figure 4A). In contrast, the skin of animals treated with Glucantime^®^ (50 mg/kg) intralesionally exhibited a focal area of inflammation in the dermis, which was characterised by the infiltration of mononuclear and polymorphonuclear cells (Figure 4E).

### 3.5. Efficacy of Topical Treatment

BALB/c mice were infected at the base of the tail, and in the fifth week of infection the experimental animals were treated topically daily with 1.7 mg of creams containing 1% or 2% of 8-HQ, blank Beeler base, and with the reference drug Glucantime^®^ by the intralesional route at the site of cutaneous lesion.

In this regard, it was observed that in the 6th week of PI (Figure 5A), the group treated with the cream containing 1% or 2% of 8-HQ had a significant decrease in the size of the skin lesions (by 34.5% and 49.5%, respectively) compared to infected animals, in which the lesions progressively increased. In the 7th week of PI, animals treated with creams containing 1 or 2% 8-HQ as well as Glucantime^®^ exhibited a significant reduction (*p* < 0.05) in the lesion size compared to infected animals. Additionally, the group of animals treated with 1% cream presented a significant decrease in the lesion size compared to groups treated with 2% cream of 8-HQ or Glucantime^®^ by the intralesional route at the site of lesion development. Lesions of animals treated with Beeler’s base cream increased over time.

Limiting the dilution assay showed that the infected group treated with Beeler’s base had the highest parasite load in the skin (Figure 5B) and lymph nodes (Figure 5C) among the studied groups. In contrast, animals treated with Glucantime^®^ by the intralesional route or topically with both creams exhibited a significative reduction in skin and lymph node parasitisms (*p* < 0.05). Although not significant, the animals treated with Glucantime^®^ showed lower parasitism than the animals treated topically.

### 3.6. Histopathological Changes

The histopathological evaluation performed in the skin histological sections of infected BALB/c mice (Figure 6A) and the group treated with blank Beeler’s base cream (Figure 6C) showed that both experimental groups had a high number of infected macrophages (detailed in insets Figure 6B,D), and this was associated with an intense and diffuse inflammatory infiltrate containing few lymphocytes, but a large number of mono and polymorphonuclear cells; furthermore, areas of necrosis were observed in the deep dermis (arrowhead).

In contrast to these data, groups treated topically with cream 1% (Figure 6E) or 2% (Figure 6G) exhibited, compared to the control, a moderate inflammatory response; additionally, it was composed mainly of lymphocytes and macrophages whose intensities of infection were lower than in the infected group (detailed in insets Figure 6F,H). Rare areas of tissue necrosis were observed. In the same way, the skin of BALB/c mice treated with Glucantime^®^ intralesionally (50 mg/kg) presented an intense and diffuse inflammatory infiltrate composed primarily of mononuclear cells (Figure 6I); also, few amastigote forms were observed within macrophages (detailed in inset Figure 6J). Focal areas of tissue necrosis were observed in the deep dermis (arrowhead in Figure 6J).

### 3.7. Immunological Studies

In the lymph nodes of the infected control and animals treated with Beeler’s base cream, a high production of IL-4 was observed. On the other hand, intralesional treatment with Glucantime^®^ and topical treatment with creams containing 1 or 2% of 8-HQ reduced IL-4 production compared to infected animals and animals treated with Beeler’s basis (*p* < 0.05). Furthermore, IL-4 levels in the groups submitted to treatments were similar to the control—uninfected, untreated animals (Figure 7A).

Concerning IFN-γ (Figure 7B), it was verified that the infected group and group treated with Beeler’s base produced low levels of IFN-γ. On the other hand, animals treated with Glucantime^®^ or cream containing 1 and 2% of 8-HQ exhibited a significant increase (*p* < 0.05) in IFN-γ production.

## 4. Discussion

8-HQ has a strong and evident anti-*Leishmania* activity against different species that cause visceral or cutaneous leishmaniasis, with selectivity indexes superior to those of the conventional drug miltefosine [10,12]. Furthermore, it was shown that the subcutaneous and intralesional administration of this molecule in mice infected with dermotropic species of leishmania drastically decreased the size of skin lesions, as well as tissue parasitism [14]. Although the therapeutic activity of 8-HQ has been described, to the best of our knowledge no topical treatment has been developed so far. The fabrication of a cream containing bioactive molecules would be a significant improvement in the treatment of cutaneous leishmaniasis because patients would be able to treat their lesions at home without the need for healthcare professionals and hospitalisation, which might be difficult in rural areas. Additionally, as suggested by the Drugs for Neglected Diseases *initiative* (DNDi), a topical treatment is a non-invasive, non-painful method of administration and possibly its use in cases of cutaneous leishmaniasis is likely to increase patient compliance and, consequently, the efficacy in comparison to current first-line treatments that are hampered by severe side effects in patients [26]. Therefore, extemporaneously or industrially manufactured topical therapies will have significative importance in geographic localities where cutaneous leishmaniasis is endemic. Thus, in the present manuscript, a cream containing the bioactive molecule 8-HQ was prepared, characterised, and assessed in terms of safety and efficacy in an animal model of cutaneous leishmaniasis.

The 1% 8-HQ cream showed a similar particle size to that of Beeler’s cream, indicating that 8-HQ is homogenously dispersed by extemporaneous dispensing and likely solubilised within the emulsion. However, the 2% 8-HQ cream showed increased sizes in D_50_ and D_90_, likely attributed to the large particle size of 8-HQ that, after incorporation in the cream, remained partially unsolubilised, resulting in suspended 8-HQ particles within the cream base. Thus, if 8-HQ remains suspended within the cream base, it might not be able to permeate across the skin and only the solubilised fraction may be able to cross the skin barrier.

Thus, the 1% cream, with more 8-HQ in solubilised form and droplets of lower particle size, was more likely to permeate across the murine and porcine skin tested compared to the 2% cream. A study of the transdermal delivery of hydroquinone demonstrated that the formulation with smaller droplets exhibited higher permeation across a Strat-M membrane and mouse skin [32].

To ensure the creams could be applied and spread on the skin, the rheology of the cream was assessed. 8-HQ creams’ shear thinned as the shear rate increased. As viscosity decreases, frictional forces also decrease, and less energy is required for shear, which occurs when a fluid is moved or spread out [37]. Measured viscosity is within the range previously approved for human use in topical formulations, and ensures that the prepared cream can be applied and stay on human skin, while the observed shear thinning behaviour ensures the spreadability of the formulation on the skin surface [36].

Creams were tested for permeability using porcine skin and an artificial membrane (Strat-M membrane). In porcine skin, 1% cream demonstrated a higher flux and diffusion coefficient through the skin compared to the 2%, with similar lag times. This is likely explained by the solubilisation of 8-HQ within the Beeler’s cream being nearly complete in the 1% compared to the 2% cream. 8-HQ has a high solubility in propylene glycol and cetyl alcohol (components of Beeler’s base), but higher amounts might be above the maximum solubility for the drug per weight of the base, which allows some of the drugs to remain suspended and thus not able to permeate across the skin. Interestingly, no difference was seen in the Strat-M membranes [34,38], which is explained by the infinite dose used in our experiments.

At the end of the permeation studies, it was observed that porcine skin treated with cream containing 1% or 2% 8-HQ accumulated 17.80 or 5.60 times more 8-HQ, respectively, than the required concentration of 8-HQ able to eliminate 50% of the number of amastigote forms (EC_50_) of *L. (L.) amazonensis*, which was previously estimated in 1.9 μg/mL after 24 h of incubation [14]. Furthermore, 8-HQ creams did not change the histological structure of porcine skin, suggesting that creams were safe to be used *in vivo*. Considering the Strat-M membrane, the diffusion coefficient of both creams was also higher than the EC_50_ of 8-HQ on *L. (L.) amazonensis* amastigotes. It is still important to note that the differences observed in the permeation of creams through the porcine skin and the Strat-M membranes may be related to the difference in the complex histological structure and thickness of the porcine skin in comparison to the artificial membrane. Nevertheless, results obtained in both experimental conditions strongly support the effectiveness of topical formulations containing 8-HQ.

During permeation experiments (*in vitro*), histological changes in the skin induced by creams containing 1 or 2% of 8-HQ were not observed. Therefore, to validate the absence of skin toxicity in the creams, non-infected BALB/c mice were treated daily for 14 days with cream 1 or 2%. In this regard, topical treatment with blank Beeler or creams containing 8-HQ was shown to not cause a histological alteration in the skin of BALB/c mice or induce an inflammatory response; in contrast, an inflammatory infiltrate was observed in the skin of BALB/c mice treated with Glucantime^®^ by the intralesional route. The results are related to the fact that Glucantime^®^ is able to increase the phagocytic capacity of monocytes and neutrophils and increase the generation of superoxide, as well as the production of TNF-α and NO [39], which explains the inflammation observed in the skin. Furthermore, animals treated by the intralesional route, which can cause an inflammatory response due to a mechanical injury, show dissociation oedema that may be responsible for this inflammatory response [40]. In summary, the results showed suggest that creams containing 1 or 2% of 8-HQ did not alter the skin morphology after 14 topical applications, and therefore they can be considered interesting topical formulations to analyse in the experimental model of cutaneous leishmaniasis.

Both creams altered the course of cutaneous lesions in mice, considering that the animals treated exhibited a significative reduction in the size of the cutaneous lesions after the first week of treatment compared to the control and the group treated only with Beeler’s cream. Additionally, 1% cream was found to induce the most effective reduction in lesion size compared to animals treated with the 2% topical cream and animals treated intralesional with Glucantime^®^ at the site of cutaneous lesion. As previously observed, this biological finding may be related to the higher permeability of 1% cream compared to the cream with 2%.

In addition to a reduction in the size of the skin lesions, a significant reduction was observed in the number of parasites in the skin and lymph nodes of infected mice submitted to topical treatment with 1 or 2% creams. Possibly, this activity may be related to the characteristics of the Beeler’s base, which contains permeability enhancers (propylene glycol, sodium lauryl sulphate, and cetyl alcohol) that aid the permeation of 8-HQ through the skin [41]. Sodium lauryl sulphate can reduce the resistance of the stratum corneum facilitating the penetration of the compound [42], and propylene glycol is a good solubiliser for hydrophobic drugs such as 8-HQ, enhancing permeation [43] through the skin, which explains the observed significant efficacy, as previously observed in mice treated with 8-HQ intralesionally [14].

Although cream and Glucatime^®^ treatments were effective in reducing the size of the lesions and parasite load, it is still important to emphasise that the animals treated by the intralesional route with Glucantime^®^ received an accumulated dose of 21 mg; such an amount reduced parasitism by 98.24 and 94.7% in the skin and lymph node, respectively. On the other hand, animals treated topically with 1% cream received a total dose of 0.24 mg of 8-HQ that decreased by 90.4 and 91.9% of parasites in the skin and lymph nodes, respectively, with comparable efficacy to Glucantime^®^ in the model of cutaneous leishmaniasis (88-fold lower dose applied).

Furthermore, it was verified that the infected control and animals treated with blank Beeler’s cream exhibited an intense and diffuse inflammatory infiltrate associated with a high number of densely infected macrophages. Areas of necrosis were observed throughout the histological section of the skin. On the other hand, the skin of animals treated with 1 or 2% creams showed a moderate inflammatory infiltrate composed mainly of lymphocytes and few infected macrophages, suggesting that the topical treatments were effective in eliminating intracellular parasites. Even though animals treated with Glucantime^®^ (50 mg/kg) demonstrated a reduced number of infected macrophages and amastigotes in tissue, histopathology revealed a significant diffuse inflammatory response, and necrosis in some areas of the skin (arrowhead in Figure 6J). It is well known that *L. (L.) amazonensis* infection induces an intense inflammatory infiltrate that is highly correlated with disease progression [44,45]. Therefore, molecules capable of decreasing this inflammatory process and reducing tissue parasitism may be of interest for the development of effective treatment, as observed in studies employing 8-HQ as an intralesional drug in the model of cutaneous leishmaniasis, which was able to reduce tissue parasitism as well as the inflammatory response [46]. Therefore, these data indicate that the action of 8-HQ formulated as a cream exerts a significant leishmanicidal effect.

During the progression of *L. (L.) amazonensis* infection, the host’s immunological response is characterised by a high production of IL-4 cytokines, resulting in a polarised Th2 immune response, leading to the survival and spread of parasites through tissues [40,47,48]. In the present study, it was verified that control animals developed a classic Th2 immune response, with high amounts of IL-4 and low amounts of IFN-γ, which is highly correlated with disease progression [49]. On the other hand, animals treated with Glucantime^®^ and creams 1 and 2% developed a Th1 immune response marked by a high production of IFN-γ and low IL-4. This polarisation has great relevance for the resistance to infection, which was confirmed by the limiting dilution assay and in the histological section of treated mice. Possibly, 8-HQ exhibits a direct action on parasites and also triggers the production of IFN-γ, which activates macrophages to a microbicidal state, leading to the death of parasites [48]. Furthermore, 8-HQ is structurally related to imidazoquinolinone, which is an agonist of the Toll-like receptor 7 (TLR-7), which has an immunomodulatory action, through the induction of interferons, and acts as an antineoplastic agent [39,50].

At the same time, a significant reduction in IL-4 levels was observed in all treated groups, indicating that the treatments were able to reverse the suppressive immune response observed in active disease [51,52], which is also important to achieve resistance [53]. In a previous study, it was observed that experimental animals with cutaneous leishmaniasis submitted to intralesional treatment with 8-HQ were also able to produce significant levels of IFN-γ [14], which reinforces that this molecule exhibits immunomodulatory activity in cutaneous leishmaniasis. However, it is still important to note that the topical administration of drugs is still an attractive option compared to the other routes of administration, especially in skin diseases, because such a type of treatment can bring the drug to the site of the diseased tissue, and it can also be administered in lower doses to develop a therapeutic effect. The peak plasma level of the drug is avoided, the bioavailability of the drug is increased due to the elimination of hepatic first-pass metabolism, and systemic toxicity is reduced, as well as leading to greater patient compliance by eliminating the need for frequent painful applications [54].

## 5. Conclusions

In conclusion, the physical characteristics of the cream demonstrated an acceptable particle size for skin permeation and creams exhibited appropriate spreadability, allowing the permeation of 8-HQ through the skin and artificial membranes. 8-HQ creams reduced the size of the lesions and the parasitic load of treated animals, as well as the inflammatory infiltrate and the density of the parasitised macrophages. This therapeutic activity was associated with an immunomodulatory activity of the molecule that stimulated the production of IFN-γ, indicating a polarised Th1 response and consequent resistance to infection. This study enables the facile extemporaneous and potentially industrial manufacture of more tolerable topical therapies for the treatment of leishmaniasis, reducing the need for hospitalisation and enabling wider access, especially to populations living in remote areas.

## Figures and Tables

**Figure 1 pharmaceutics-15-02602-f001:**
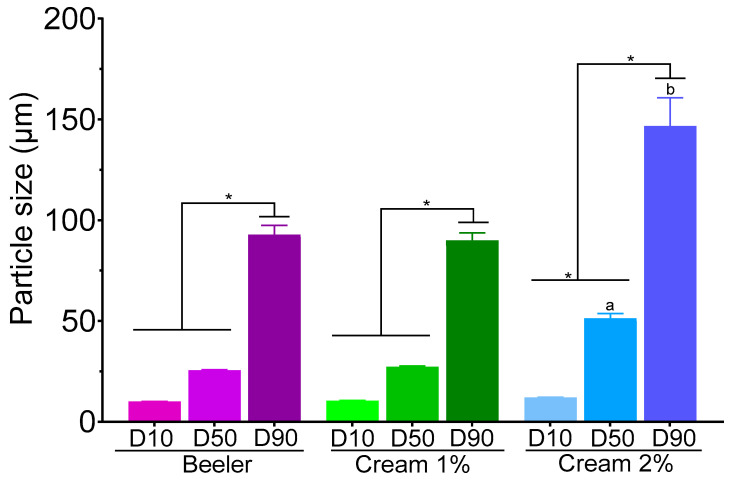
Particle size distribution of Beeler’s base and creams containing 1 or 2% of 8-HQ. Data are expressed as mean ± standard deviation of three independent experiments. * *p* < 0.05 indicates statistical significance between the indicated groups. The letters “a” and “b” indicate statistical significance among D_50_ and D_90_ of cream 2% with Beeler base and cream 1%, respectively.

**Figure 2 pharmaceutics-15-02602-f002:**
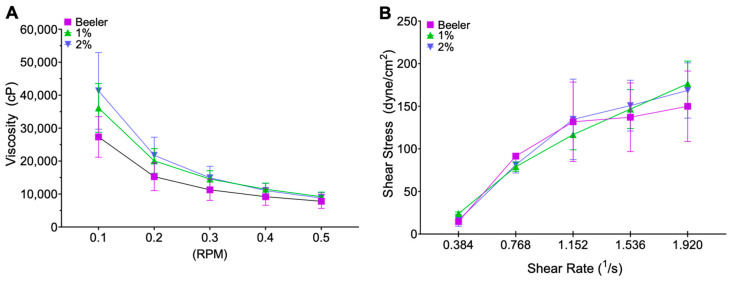
Viscosity versus applied rpm (**A**), from 0.5 to 0 rpm. Blank Beeler’s base cream, 1%, and 2% 8-HQ cream. Shear stress versus shear rate (**B**), from 0.384 to 1.92 (1/s)). Blank Beeler’s base cream with a y = 82.378x + 10.07 and a R^2^ = 0.8257, 1% 8-HQ cream with a y = 97.017x − 3.18 and a R^2^ = 0.9808, 2% 8-HQ cream with a y = 97.404x − 1.9483 and a R^2^ = 0.914.

**Figure 3 pharmaceutics-15-02602-f003:**
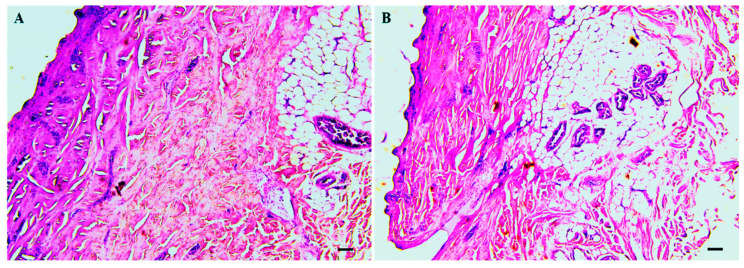
After the skin permeation test, fragments of porcine ear skin were shaved and fixed in 2% buffered formalin and the histological sections were stained with haematoxylin and eosin. (HE). (**A**) Cream containing 1% 8-HQ; (**B**) cream containing 2% 8-HQ. The black bar represents 10 µm.

**Figure 4 pharmaceutics-15-02602-f004:**
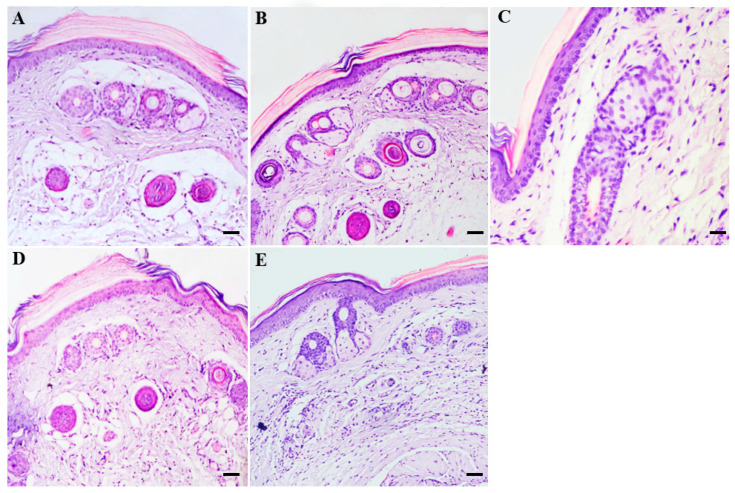
Histological changes in non-infected BALB/c mice submitted to the treatment. Control mice were not submitted to any topical treatment (**A**); on the other side, groups of mice were treated for 14 days with blank Beeler’s base (**B**) or creams containing 1% (**C**) or 2% of 8-HQ (**D**) topically or with Glucantime^®^ (50 mg/kg) by the intralesional route (**E**). The skin histological section of each group was stained with haematoxylin and eosin (HE) and the main histological changes were compared with the skin histological section collected from untreated BALB/c mice (**A**). Bars = 20 µm.

**Figure 5 pharmaceutics-15-02602-f005:**
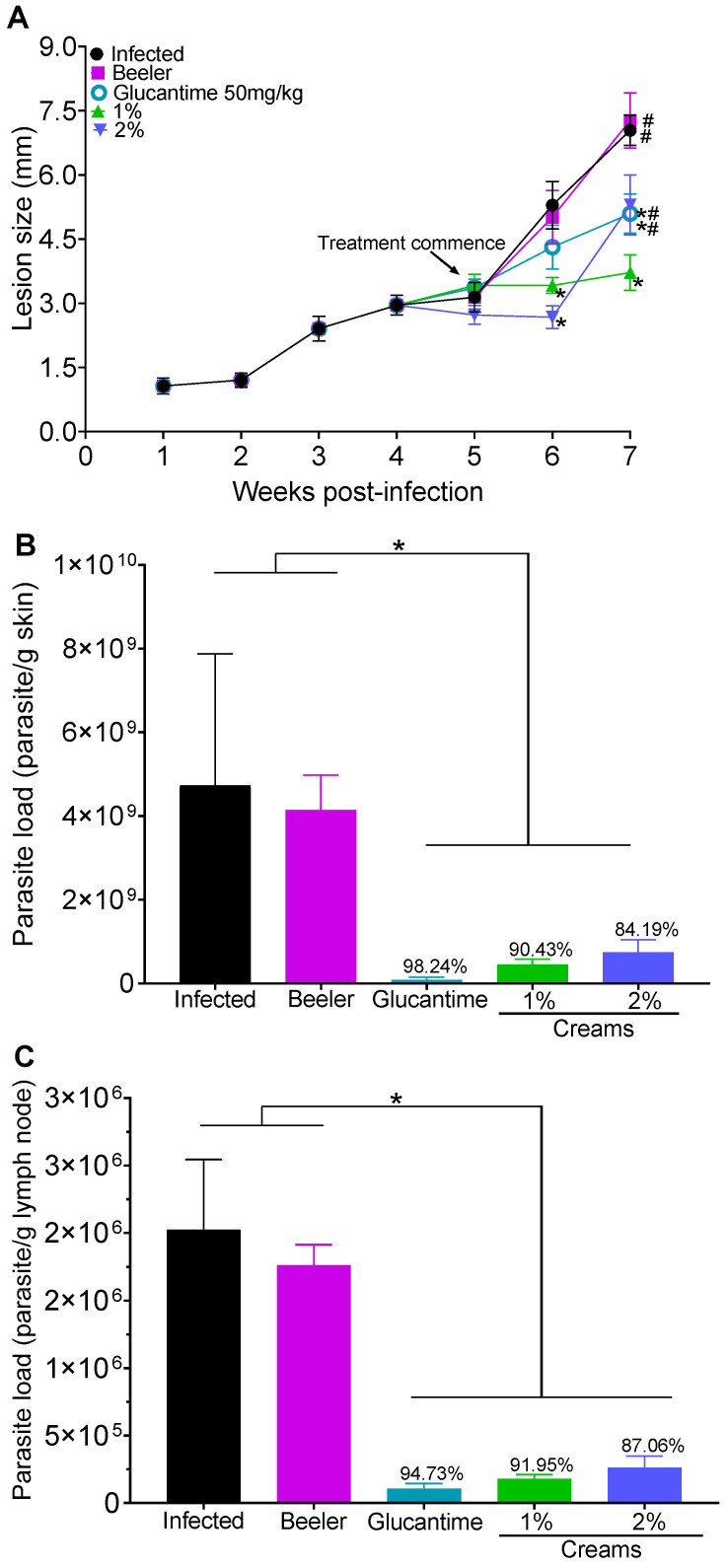
BALB/c mice were infected with 10^6^ promastigote forms of *L (L.) amazonensis* at the base of the tail; at the 5th week post-infection, topical treatment was started with creams containing 1 or 2% of 8-HQ or Glucantime^®^ (50 mg/kg) by the intralesional route at the site of cutaneous lesion. The animals were treated once a day for 14 days. Lesion development (**A**) was monitored with a micrometer and at the 7th week PI parasite loads in the skin (**B**) and lymph nodes (**C**) were determined using a limiting dilution assay. * *p* < 0.05 in comparison to the infected group. # *p* < 0.05 in comparison to the 1% cream.

**Figure 6 pharmaceutics-15-02602-f006:**
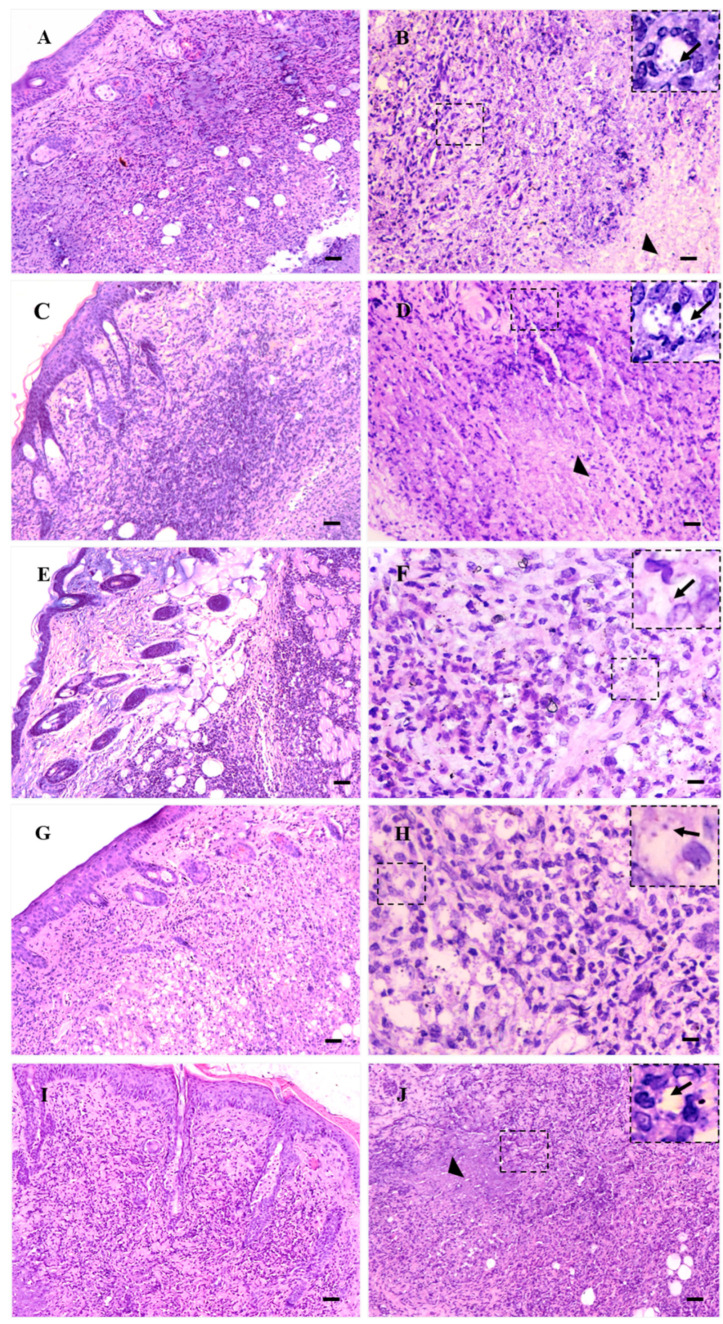
Histological skin sections of BALB/c mice infected with *L. (L.) amazonensis*. (**A**,**B**) Infected control; (**C**,**D**) topical treatment with blank Beeler’s base cream; (**E**,**F**) topical treatment with cream 1% of 8-hydroxyquinoline; (**G**,**H**) topical treatment with cream 2% of 8-hydroxyquinoline; (**I**,**J**) treatment by the intralesional route at the site of cutaneous lesion with 50 mg/kg of Glucantime^®^. (**A**,**C**,**E**,**G**,**I**) show details of the epidermis and dermis of each group, while (**B**,**D**,**F**,**H**,**J**) show details mainly on parasitism, illustrated by the black arrows. Arrowheads indicate areas of necrosis (bars: 40 μm).

**Figure 7 pharmaceutics-15-02602-f007:**
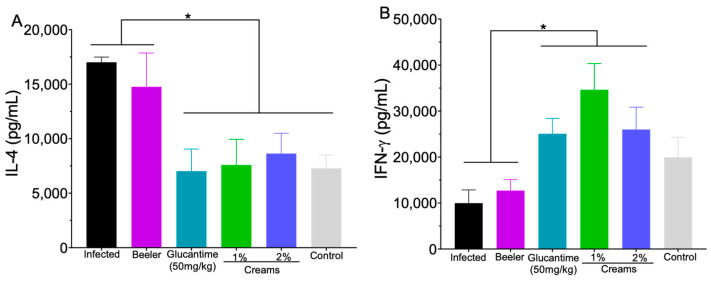
The draining lymph nodes of all experimental animals were collected, a single cell suspension was produced, and the supernatants were collected to quantify the amounts of IL-4 (**A**) and IFN-γ (**B**) using ELISA. * *p* < 0.05.

**Table 1 pharmaceutics-15-02602-t001:** Transdermal diffusion tests were performed with porcine skin and artificial membranes incubated with creams containing 1 or 2% of 8-HQ. * *p* < 0.05 between 1 and 2% creams.

	Porcine Skin		Artificial Membrane	
Parameter	Cream 1%	Cream 2%	Cream 1%	Cream 2%
Steady-state flux (μg/cm^2^/h)	40.24 ± 1.09	25.55 ± 1.21 *	30.66 ± 3.25	33.49 ± 6.09
Lag time (h)	0.46 ± 0.03	0.60 ± 0.16	0.35 ± 0.03	0.24 ± 0.05
Permeability coefficient (mm^2^/h)	0.20 ± 0.02	0.06 ± 0.003 *	0.31 ± 0.03	0.24 ± 0.03
Diffusion coefficient (μm/h)	33.83 ± 4.68	10.63 ± 0.50 *	9.20 ± 0.97	7.33 ± 0.87
Cumulative amount of 8-HQ retained (μg/mg)	0.20 ± 0.02	0.17 ± 0.04	0.10 ± 0.01	0.12 ± 0.01

## Data Availability

The data presented in this study are available in this article.

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
