# Peer review of "Therapeutic Activity of a Topical Formulation Containing 8-Hydroxyquinoline for Cutaneous Leishmaniasis"

_pharmaceutics, 2023, doi:10.3390/pharmaceutics15112602_

Round 1

Reviewer 1 Report

Comments and Suggestions for Authors

I have carefully reviewed the paper titled "Therapeutic activity of a topical formulation containing 8-2 hydroxyquinoline for cutaneous leishmaniasis". Data is presented well, however, there are many major issues, which are unclear in current draft. Aiming to improve the quality of the paper my comments are given below.

1.      Line 156. Creams were packaged in bottles of jars?

2.      Line 158 in Materials and methods section please emphasize on the method used not the instrument.

3.      Lines 154-155 “The active compound (8-HQ at 1% and 2%) was dispersed in 50 μL of propylene glycol prior being incorporated into the Beeler’s base using a mortar and pestle”.

How much 8-HQ was dispersed in each case in propylene glycol and in what mass of cream was the mixture incorporated. 50μL seems like a very small amount for dispersing 8-HQ.

Why a pestle and mortar? Do the authors aim to prepare the samples extemporaneously or industrially? They state that they are “ paving the way towards a novel industrially scalable and readily translatable topical treatment” (line 135) but clearly the preparation method used is not scalable unless substantial modifications were achieved.

If the formulations were prepared following a different technique like homogenization at the end of cream preparation and API incorporation do the authors estimate that 2% 8-HQ would be incorporated in a cream with smaller droplet size and the results would have been different?

Use of co-surfactants, some solubilizing agent or even a different galenic base may result in more efficient formulations. Why was this specific cream base chosen? What are its advantages compared to other available cream bases?

4.      Line 205. Parenthesis is not needed.

5.      Line 211. Standard curves are constructed or plotted. Not built.

6.      Lines 219-233. All components of the equations and their units should be explained in the text following each equation.

7.      Line 315. The number of the figure needs to be corrected.

8.      Line 356 and 361 authors use the term “intradermally / intradermic” but in other parts of the text they use the term “intralesional”. Only one term should be used for text consistency.

9.      Figure 4 caption. The Image numbering should be presented in alphabetical order.

10.   Lines 354-355 cream containing 1% (Figure 4C), or 2% (Figure 4D) of 8-HQ showed similar morphology to the control group (Figure 4A).

Looking at both C and D there is an obviously thinner epidermis of the animals treated with both creams (containing 1% and 2% of 8-HQ ) compared to the controls (A or B). In E this thinning is not present. How do the authors explain this? Why do they not comment on this finding?

11.   Several terms are used in different ways throughout the text. Some examples follow:

·          The way that glucantime is mentioned throughout the text is not uniform. It is either Glucantime® or glucantime.

·          Line 276 digital micrometer is mentioned but then in line 389 is referred as caliper.

·          The treatment route of Glucantime® is described as intralesional and subcutaneous route (Line 498 502).

·           

12.   The results of all Histological tests with relevant figures on porcine skin or mice skin should be grouped under one section title (maybe divided by distinctive subtitles). It also should be mentioned after section title 2.5. Efficacy of topical treatment.

13.   The numbering of Section title 2.5. Efficacy of topical treatment should be corrected.

14.   Lines 374-378 The paragraph needs rephrasing for clarity.

15.   Line 416 “…head of the arrows head”. Do the authors mean “point of the arrowheads”?

16.   The cream was prepared not produced as stated throughout the text (example: Lines153, 439, 466, 447)

17.   Please check the text for language use errors. Some examples follow:

·                   Line 502 Furthermore, such animals were treated…

·                  Line 504 In summary, such results…

18.    In general it would be misleading to claim that cream with 1% 8 HQ was better than the respective cram with 2% 8-HQ, since the latter was not correctly formulated because the active ingredient was not properly dissolved before incorporation in the cream base, according to authors.

19.   An image of the creams with a simple optical microscope would reveal the potential presence of crystals of non-dissolved 8 HQ.

20.   Line 580 a conclusion (not a summary) is needed here.

Comments on the Quality of English Language

  The text needs extensive review on the proper use of words (plesae see comment # 17. Authors should also avoid repetition of same phrases (like: on the other hand).

Author Response

Dear Reviewer,

We thank the reviewer 1  for the comments and we have carefully reviewed the manuscript to address the comments. Changes are highlighted in red.

Please, see attached file.

Reviewer 2 Report

Comments and Suggestions for Authors

1.     Please indicate the greatest innovation of the research conducted and the results obtained

2.     There is no information about the permeability of the pure substance, as well as a positive standard

3.     Conclusions part is missing

Author Response

Reviewer 2.

We thank the reviewer for the comments and we have carefully reviewed the manuscript to address the comments. Changes are highlighted in purple as shown below.

  1. Please indicate the greatest innovation of the research conducted and the results obtained

This is highlighted at the end of the introduction section.

  1. There is no information about the permeability of the pure substance, as well as a positive standard

We have not compared the permeability of 8-HQ dissolved in an organic solubiliser as these would not be applicable to human or animal application, but we agree that would provide useful information regarding the flux of the 8-HQ. There is no commercial formulation of 8-HQ that we can compare against.

3.Conclusions part is missing

We have added conclusions as per reviewer’s request.

Reviewer 3 Report

Comments and Suggestions for Authors

8-Hydroxyquinoline (8-HQ) and some of its derivatives have already been described as potent antileishmanial compounds, in vitro on various Leishmania species but also in vivo. In a previous publication (Pharmaceuticals 2023), some of the authors of the present manuscript reported the efficacy of 8-HQ administered by intralesional route on L amazonensis infected mice. In this work, S.K. Santos de Lima et al. study the effect of a topical formulation containing 1 or 2% of 8-HQ to treat cutaneous leishmaniasis.

The effect of the formulations was first evaluated on different skin models. Then the effect on parasite burden and lesion size reduction in BALB/c mice infected with L amazonensis was measured, using glucantime (intralesional administration) as control. Histopathologic changes upon treatment and immunological studies (IL-4 and IFN-gamma) were also considered.

The experiments support the conclusions regarding the effect of the 1- or 2 % 8-HQ topical creams.

However, I have a few questions and remarks:

-        The advantages of using topical cream compared to classical treatment of cutaneous leishmaniosis, including intralesional administration, is clearly described in the discussion part. However, in that paper, the authors chose to compare their topical formulation to intralesional glucantime. It would have been interesting to also have a control with the 8-HQ intralesional treatment.

-        Have stability assays been conducted on the 8-HQ creams?

In addition, two minor points have to be corrected:

1.               The description of the 8-hydroxyquinoline is trivial. In consequence, “a bicyclic heterocyclic scaffold that contains a phenol ring fused with a pyridine ring at two adjacent carbons” should be removed.

2.               P 11 line 409: replace “arrow red” by “arrow head”

I can recommend publication provided the authors respond to the remarks that I outlined above.

Author Response

We thank the reviewer for the comments and we have carefully reviewed the manuscript to address the comments. Changes are highlighted in green as shown below.

-        The advantages of using topical cream compared to classical treatment of cutaneous leishmaniosis, including intralesional administration, is clearly described in the discussion part. However, in that paper, the authors chose to compare their topical formulation to intralesional glucantime. It would have been interesting to also have a control with the 8-HQ intralesional treatment.

            Glucantime® intralesionally is the standard of care for patients. The aim of our study was to develop non-invasive and tolerable medicines that can be administered by the patient without the need for hospitalization and this is why we focus on presenting the topical administration results. We have recently published the effect of 8-HQ intralesionally (10mg/kg) vs Glucantime ® (50mg/kg) (Pharmaceuticals (Basel). 2023 May; 16(5): 707, doi: 10.3390/ph16050707).

-        Have stability assays been conducted on the 8-HQ creams?

            We have not undertaken stability studies for the 8-HQ creams. Visual observations over a period of a month indicated similar appearance and consistency, but at this moment of time, we don’t have stability data to ensure product shelf-life post extemporaneous manufacture further than a month.

In addition, two minor points have to be corrected:

  1. The description of the 8-hydroxyquinoline is trivial. In consequence, “a bicyclic heterocyclic scaffold that contains a phenol ring fused with a pyridine ring at two adjacent carbons” should be removed.

            This section was removed as suggested.

  1. P 11 line 409: replace “arrow red” by “arrow head”

            We apologize to the reviewer and amended appropriately.

Round 2

Reviewer 1 Report

Comments and Suggestions for Authors

Τhe authors have  adequately answered all the points raised. 

Comments on the Quality of English Language

The authors have corrected several point of the manuscript. Nevertheless the text still needs a careful edit for syntax errors (see some examples at  lines 454, 512 etc).

Reviewer 2 Report

Comments and Suggestions for Authors

Accept in present form